# HOW TO EXPLOIT HYPERSPHERICAL EMBEDDINGS FOR OUT-OF-DISTRIBUTION DETECTION?

**Yifei Ming[1], Yiyou Sun[1], Ousmane Dia[2], Yixuan Li[1]**
Department of Computer Sciences, University of Wisconsin-Madison[1]
Meta[2]
{alvinming,sunyiyou,sharonli}@cs.wisc.edu, ousamdia@meta.com

## ABSTRACT

Out-of-distribution (OOD) detection is a critical task for reliable machine learning. Recent advances in representation learning give rise to distance-based OOD detection, where testing samples are detected as OOD if they are relatively far away from the centroids or prototypes of in-distribution (ID) classes. However, prior methods directly take off-the-shelf contrastive losses that suffice for classifying ID samples, but are not optimally designed when test inputs contain OOD samples. In this work, we propose CIDER, a novel representation learning framework that exploits hyperspherical embeddings for OOD detection. CIDER jointly optimizes two losses to promote strong ID-OOD separability: a dispersion loss that promotes large angular distances among different class prototypes, and a compactness loss that encourages samples to be close to their class prototypes. We analyze and establish the unexplored relationship between OOD detection performance and the embedding properties in the hyperspherical space, and demonstrate the importance of dispersion and compactness. CIDER establishes superior performance, outperforming the latest rival by 13.33% in FPR95. Code is available at https://github.com/deeplearning-wisc/cider.

## 1  INTRODUCTION

When deploying machine learning models in the open world, it is important to ensure the reliability of the model in the presence of out-of-distribution (OOD) inputs—samples from an unknown distribution that the network has not been exposed to during training, and therefore should not be predicted with high confidence at test time. We desire models that are not only accurate when the input is drawn from the known distribution, but are also aware of the unknowns outside the training categories. This gives rise to the task of OOD detection, where the goal is to determine whether an input is in-distribution (ID) or not.

A plethora of OOD detection algorithms have been developed recently, among which distance-based methods demonstrated promise (Lee et al., 2018; Xing et al., 2020). These approaches circumvent the shortcoming of using the model's confidence score for OOD detection, which can be abnormally high on OOD samples (Nguyen et al., 2015) and hence not distinguishable from ID data. Distance-based methods leverage feature embeddings extracted from a model, and operate under the assumption that the test OOD samples are relatively far away from the clusters of ID data.

Arguably, the efficacy of distance-based approaches can depend largely on the quality of feature embeddings. Recent works including SSD+ (Sehwag et al., 2021) and KNN+ (Sun et al., 2022) directly employ off-the-shelf contrastive losses for OOD detection. In particular, these works use the supervised contrastive loss (SupCon) (Khosla et al., 2020) for learning the embeddings, which are then used for OOD detection with either parametric Mahalanobis distance (Lee et al., 2018; Sehwag et al., 2021) or non-parametric KNN distance (Sun et al., 2022). However, existing training objectives produce embeddings that suffice for classifying ID samples, but remain sub-optimal for OOD detection. For example, when trained on CIFAR-10 using SupCon loss, the average angular distance between ID and OOD data is only 29.86 degrees in the embedding space, which is too small for effective ID-OOD separation. This raises the important question:

*How to exploit representation learning methods that maximally benefit OOD detection?*

In this work, we propose CIDER, a **C**ompactness and **D**isp**E**rsion **R**egularized learning framework designed for OOD detection. Our method is motivated by the desirable properties of hyperspherical embeddings, which can be naturally modeled by the von Mises-Fisher (vMF) distribution. vMF is a classical and important distribution in directional statistics (Mardia et al., 2000), is analogous to spherical Gaussian distributions for features with unit norms. Our key idea is to design an end-to-end trainable loss function that enables optimizing hyperspherical embeddings into a mixture of vMF distributions, which satisfy two properties simultaneously: **(1)** each sample has a higher probability assigned to the correct class in comparison to incorrect classes, and **(2)** different classes are far apart from each other. To formalize our idea, CIDER introduces two losses: a *dispersion loss* that promotes large angular distances among different class prototypes, along with a *compactness loss* that encourages samples to be close to their class prototypes. These two terms are complementary to shape hyperspherical embeddings for both OOD detection and ID classification purposes. Unlike previous contrastive loss, CIDER explicitly formalizes the latent representations as vMF distributions, thereby providing a direct theoretical interpretation of hyperspherical embeddings.

In particular, we show that promoting large inter-class dispersion is key to strong OOD detection performance, which has not been explored in previous literature. Previous methods including SSD+ directly use off-the-shelf SupCon loss, which produces embeddings that lack sufficient inter-class dispersion needed for OOD detection. CIDER mitigates the issue by explicitly optimizing for large inter-class margins and leads to more desirable hyperspherical embeddings. Noticeably, when trained on CIFAR-10, CIDER displays a relative 42.36% improvement of ID-OOD separability compared to SupCon. We further show that CIDER's strong representation can benefit different distance-based OOD scores, outperforming recent competitive methods SSD+ (Sehwag et al., 2021) and KNN+ (Sun et al., 2022) by a significant margin. Our key results and contributions are:

1. We propose CIDER, a novel representation learning framework designed for OOD detection. Compared to the latest rival (Sun et al., 2022), CIDER produces superior embeddings that lead to **13.33%** error reduction (in FPR95) on the challenging CIFAR-100 benchmark.

2. We are the first to establish the unexplored relationship between OOD detection performance and the embedding quality in the hyperspherical space, and provide measurements based on the notion of compactness and dispersion. This allows future research to quantify the embedding in the hyperspherical space for effective OOD detection.

3. We offer new insights on the design of representation learning for OOD detection. We also conduct extensive ablations to understand the efficacy and behavior of CIDER, which remains effective and competitive under various settings, including the ImageNet dataset.

## 2 PRELIMINARIES

We consider multi-class classification, where $\mathcal{X}$ denotes the input space and $\mathcal{Y}^{\text{in}} = \{1, 2, ..., C\}$ denotes the ID labels. The training set $\mathcal{D}_{\text{tr}}^{\text{in}} = \{(\mathbf{x}_i, y_i)\}_{i=1}^{N}$ is drawn *i.i.d.* from $P_{\mathcal{X}\mathcal{Y}^{\text{in}}}$. Let $P_{\mathcal{X}}$ denote the marginal distribution over $\mathcal{X}$, which is called the in-distribution (ID).

**Out-of-distribution detection.** OOD detection can be viewed as a binary classification problem. At test time, the goal of OOD detection is to decide whether a sample $\mathbf{x} \in \mathcal{X}$ is from $P_{\mathcal{X}}$ (ID) or not (OOD). In practice, OOD is often defined by a distribution that simulates unknowns encountered during deployment, such as samples from an irrelevant distribution whose label set has no intersection with $\mathcal{Y}^{\text{in}}$ and therefore should not be predicted by the model. Mathematically, let $\mathcal{D}_{\text{test}}^{\text{ood}}$ denote an OOD test set where the label space $\mathcal{Y}^{\text{ood}} \cap \mathcal{Y}^{\text{in}} = \emptyset$. The decision can be made via a level set estimation: $G_\lambda(\mathbf{x}) = \mathbb{1}\{S(\mathbf{x}) \geq \lambda\}$, where samples with higher scores $S(\mathbf{x})$ are classified as ID and vice versa. The threshold $\lambda$ is typically chosen so that a high fraction of ID data (*e.g.* 95%) is correctly classified.

**Hyperspherical embeddings.** A hypersphere is a topological space that is homeomorphic to a standard $n$-sphere, which is the set of points in $(n+1)$-dimensional Euclidean space that are located at a constant distance from the center. When the sphere has a unit radius, it is called the unit hypersphere. Formally, an $n$-dimensional unit-hypersphere $S^n := \{\mathbf{z} \in \mathbb{R}^{n+1} | \|\mathbf{z}\|_2 = 1\}$. Geometrically, hyperspherical embeddings lie on the surface of a hypersphere.

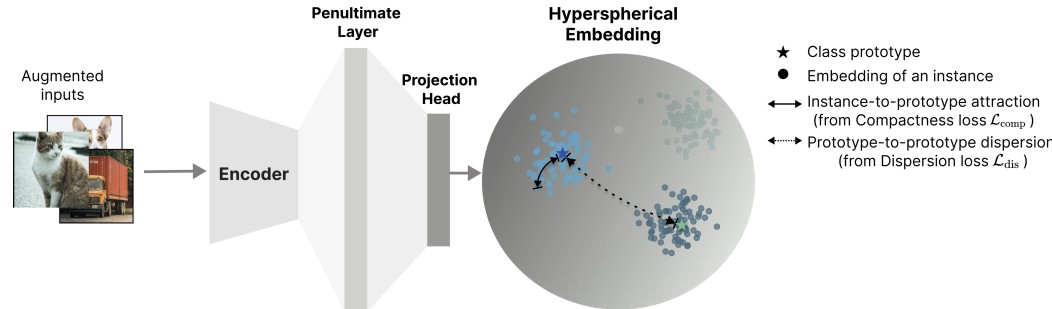

Figure 1: Overview of our compactness and dispersion regularized (**CIDER**) learning framework for OOD detection. We jointly optimize two complementary terms to encourage desirable properties of the embedding space: (1) a *dispersion loss* to encourage larger angular distances among different class prototypes, and (2) a *compactness loss* to encourage samples to be close to their class prototypes.

## 3 METHOD

**Overview.** Our framework CIDER is illustrated in Figure 1. The general architecture consists of two components: (1) a deep neural network encoder $f : \mathcal{X} \mapsto \mathbb{R}^e$ that maps the augmented input $\tilde{\mathbf{x}}$ to a high dimensional feature embedding $f(\tilde{\mathbf{x}})$ (often referred to as the penultimate layer features); (2) a projection head $h : \mathbb{R}^e \mapsto \mathbb{R}^d$ that maps the high dimensional embedding $f(\tilde{\mathbf{x}})$ to a lower dimensional feature representation $\tilde{\mathbf{z}} := h(f(\tilde{\mathbf{x}}))$. The loss is applied to the normalized feature embedding $\mathbf{z} := \tilde{\mathbf{z}}/\|\tilde{\mathbf{z}}\|_2$. The normalized embeddings are also referred to as *hyperspherical embeddings*, since they are on a unit hypersphere. Our goal is to shape the hyperspherical embedding space so that the learned embeddings can be mostly effective for distinguishing ID vs. OOD data.

### 3.1 MODEL HYPERSPHERICAL EMBEDDINGS

The hyperspherical embeddings can be naturally modeled by the von Mises-Fisher (vMF) distribution, a classical and important distribution in directional statistics (Mardia et al., 2000). In particular, vMF is analogous to spherical Gaussian distributions for features $\mathbf{z}$ with unit norms ($\|\mathbf{z}\|^2 = 1$). The probability density function for a unit vector $\mathbf{z} \in \mathbb{R}^d$ in class $c$ is defined as:

$$p_d(\mathbf{z}; \boldsymbol{\mu}_c, \kappa) = Z_d(\kappa) \exp\left(\kappa \boldsymbol{\mu}_c^\top \mathbf{z}\right),$$
(1)

where $\boldsymbol{\mu}_c$ is the class prototype with unit norm, $\kappa \geq 0$ indicates the tightness of the distribution around the mean direction $\boldsymbol{\mu}_c$, and $Z_d(\kappa)$ is the normalization factor. The larger value of $\kappa$, the stronger the distribution is concentrated in the mean direction. In the extreme case of $\kappa = 0$, the sample points are distributed uniformly on the hypersphere.

Under this probability model, an embedding vector $\mathbf{z}$ is assigned to class $c$ with the following normalized probability:

$$\mathbb{P}(y = c|\mathbf{z}; \{\kappa, \boldsymbol{\mu}_j\}_{j=1}^C) = \frac{Z_d(\kappa) \exp\left(\kappa \boldsymbol{\mu}_c^\top \mathbf{z}\right)}{\sum_{j=1}^C Z_d(\kappa) \exp\left(\kappa \boldsymbol{\mu}_j^\top \mathbf{z}\right)}$$
(2)

$$= \frac{\exp\left(\boldsymbol{\mu}_c^\top \mathbf{z}/\tau\right)}{\sum_{j=1}^C \exp\left(\boldsymbol{\mu}_j^\top \mathbf{z}/\tau\right)},$$
(3)

where $\kappa = \frac{1}{\tau}$. Next, we outline our proposed training method that promotes class-conditional vMF distributions for OOD detection.

### 3.2 HOW TO OPTIMIZE HYPERSPHERICAL EMBEDDINGS?

**Training objective.** Our key idea is to design a trainable loss function that enables optimizing hyperspherical embeddings into a mixture of vMF distributions, which satisfy two properties simultaneously: **(1)** each sample has a higher probability assigned to the correct class in comparison to incorrect classes, and **(2)** different classes are far apart from each other.

To achieve ①, we can perform maximum likelihood estimation (MLE) on the training data:

$$\text{argmax}_\theta \prod_{i=1}^{N} p(y_i|\mathbf{z}_i; \{\kappa_j, \boldsymbol{\mu}_j\}_{j=1}^{C}), \tag{4}$$

where $i$ is the index of the embedding and $N$ is the size of the training set. By taking the negative log-likelihood, the objective function is equivalent to minimizing the following loss:

$$\mathcal{L}_{\text{comp}} = -\frac{1}{N} \sum_{i=1}^{N} \log \frac{\exp\left(\mathbf{z}_i^\top \boldsymbol{\mu}_{c(i)}/\tau\right)}{\sum_{j=1}^{C} \exp\left(\mathbf{z}_i^\top \boldsymbol{\mu}_j/\tau\right)}, \tag{5}$$

where $c(i)$ denotes the class index of a sample $\mathbf{x}_i$, and $\tau$ is the temperature parameter. We term this *compactness loss*, since it encourages samples to be closely aligned with its class prototype.

To promote property ②, we propose the *dispersion loss*, optimizing large angular distances among different class prototypes:

$$\mathcal{L}_{\text{dis}} = \frac{1}{C} \sum_{i=1}^{C} \log \frac{1}{C-1} \sum_{j=1}^{C} \mathbb{1}\{j \neq i\} e^{\boldsymbol{\mu}_i^\top \boldsymbol{\mu}_j/\tau}. \tag{6}$$

While prototypes with larger pairwise angular distances may not impact ID classification accuracy, they are crucial for OOD detection, as we will show later in Section 4.2. Since the embeddings of OOD samples lie in-between ID clusters, optimizing large inter-class margin benefits OOD detection. The importance of inter-class dispersion can also be explained in Figure 2, where samples in the `fox` class (OOD) are semantically close to `cat` (ID) and `dog` (ID). A larger angular distance (*i.e.* smaller cosine similarity) between ID classes `cat` and `dog` in the hyperspherical space improves the separability from `fox`, and allows for more effective detection. We investigate this phenomenon quantitatively in Section 4.

Figure 2: Illustration of desirable hyperspherical embeddings for OOD detection. As OOD samples lie *between* ID clusters, optimizing a large angular distance among ID clusters benefits OOD detection.

Formally, our training objective **CIDER** (compactness and dispersion regularized learning) is:

$$\mathcal{L}_{\text{CIDER}} = \mathcal{L}_{\text{dis}} + \lambda_c \mathcal{L}_{\text{comp}}, \tag{7}$$

where $\lambda_c$ is the co-efficient modulating the relative importance of two losses. These two terms are complementary to shape hyperspherical embeddings for both ID classification and OOD detection.

**Prototype estimation and update.** During training, an important step is to estimate the class prototype $\boldsymbol{\mu}_c$ for each class $c \in \{1, 2, ..., C\}$. One canonical way to estimate the prototypes is to use the mean vector of all training samples for each class and update it frequently during training. Despite its simplicity, this method incurs a heavy computational toll and causes undesirable training latency. Instead, the class-conditional prototypes can be effectively updated in an exponential-moving-average manner (EMA) (Li et al., 2020):

$$\boldsymbol{\mu}_c := \text{Normalize}(\alpha \boldsymbol{\mu}_c + (1-\alpha)\mathbf{z}), \ \forall c \in \{1, 2, \ldots, C\} \tag{8}$$

where the prototype $\boldsymbol{\mu}_c$ for class $c$ is updated during training as the moving average of all embeddings with label $c$, and $\mathbf{z}$ denotes the normalized embedding of samples of class $c$. We ablate the effect of prototype update factor $\alpha$ in Section C. The pseudo-code for our method is in Appendix A.

**Remark 1: Differences *w.r.t.* SSD+.** We highlight three fundamental differences *w.r.t.* SSD+, in terms of training objective, test-time OOD detection, and theoretical interpretation. **(1)** At training time, SSD+ directly uses off-the-shelf SupCon loss (Khosla et al., 2020), which produces embeddings that *lack sufficient inter-class dispersion needed for OOD detection*. For example, when trained on CIFAR-10 using SupCon loss, the average angular distance between ID and OOD data is only 29.86 degrees in the embedding space. In contrast, CIDER enforces the inter-class dispersion by *explicitly* maximizing the angular distances among different ID class prototypes. As we

will show in Section 4.2, CIDER displays a relative 42.36% improvement of ID-OOD separability compared to SSD+, due to the explicit inter-class dispersion. **(2)** At test time, SSD+ uses the Mahalanobis distance (Lee et al., 2018), which imposes a strong Gaussian distribution assumption on hyperspherical embeddings. In contrast, CIDER alleviates this assumption with a non-parametric distance score. **(3)** Lastly, CIDER explicitly models the latent representations as vMF distributions, providing a direct and clear geometrical interpretation of hyperspherical embeddings.

**Remark 2: Differences *w.r.t.* Proxy-based methods.** Our work also bears significant differences *w.r.t.* proxy-based metric learning methods. (1) Our primary task is OOD detection, whereas deep metric learning is commonly used for face verification and image retrieval tasks; (2) Prior methods such as ProxyAnchor (Kim et al., 2020) lack explicit prototype-to-prototype dispersion, which we show is crucial for OOD detection. Moreover, ProxyAnchor initializes the proxies randomly and updates through gradients, while we estimate prototypes directly from sample embeddings using EMA. We provide experimental comparisons next.

# 4 EXPERIMENTS

## 4.1 COMMON SETUP

**Datasets and training details.** Following the common benchmarks in the literature, we consider CIFAR-10 and CIFAR-100 (Krizhevsky et al., 2009) as in-distribution datasets. For OOD test datasets, we use a suite of natural image datasets including SVHN (Netzer et al., 2011), Places365 (Zhou et al., 2017), Textures (Cimpoi et al., 2014b), LSUN (Yu et al., 2015), and iSUN (Xu et al., 2015). In our main experiments, we use ResNet-18 as the backbone for CIFAR-10 and ResNet-34 for CIFAR-100. We train the model using stochastic gradient descent with momentum 0.9, and weight decay $10^{-4}$. To demonstrate the simplicity and effectiveness of CIDER, we adopt the **same** hyperparameters as in SSD+ (Sehwag et al., 2021) with the SupCon loss: the initial learning rate is 0.5 with cosine scheduling, the batch size is 512, and the training time is 500 epochs. We choose the default weight $\lambda_c = 2$, so that the value of different loss terms are similar upon model initialization. Following the literature (Khosla et al., 2020), we use the embedding dimension of 128 for the projection head. The temperature $\tau$ is 0.1. We adjust the prototype update factor $\alpha$, batch size, temperature, loss weight, prototype update factor, and model architecture in our ablation study (Appendix C). We report the ID classification results for SSD+ and CIDER following the common linear evaluation protocol (Khosla et al., 2020), where a linear classifier is trained on top of the normalized penultimate layer features. More experimental details are provided in Appendix B. Code and data is released publicly for reproducible research.

**OOD detection scores.** During test time, we employ a distance-based method for OOD detection. An input **x** is considered OOD if it is relatively far from the ID data in the embedding space. By default, we adopt a simple non-parametric KNN distance (Sun et al., 2022), which does not impose any distributional assumption on the feature space. Here the distance is the cosine similarity with respect to the $k$-th nearest neighbor, which is equivalent to the (negated) Euclidean distance as all features have unit norms. In the ablation study, we also consider the commonly used Mahalanobis score (Lee et al., 2018) for a fair comparison with SSD+ (Sehwag et al., 2021).

**Evaluation metrics.** We report the following metrics: (1) the false positive rate (FPR95) of OOD samples when the true positive rate of ID samples is at 95%, (2) the area under the receiver operating characteristic curve (AUROC), and (3) ID classification accuracy (ID ACC).

## 4.2 MAIN RESULTS AND ANALYSIS

**CIDER outperforms competitive approaches.** Table 1 contains a wide range of competitive methods for OOD detection. All methods are trained on ResNet-34 using CIFAR-100, without assuming access to auxiliary outlier datasets. For clarity, we divide the methods into two categories: trained with and without contrastive losses. For pre-trained model-based scores such as MSP (Hendrycks & Gimpel, 2017), ODIN (Liang et al., 2018), Mahalanobis (Lee et al., 2018), and Energy (Liu et al., 2020), the model is trained with the softmax cross-entropy (CE) loss. GODIN (Hsu et al., 2020) is trained using the DeConf-C loss, while LogitNorm (Wei et al., 2022) modifies CE loss via logit normalization. For methods involving contrastive losses, we consider ProxyAnchor (Kim et al., 2020), SimCLR (Winkens et al., 2020), CSI (Tack et al., 2020), SSD+ (Sehwag et al., 2021), and KNN+ (Sun et al., 2022). Both SSD+ and KNN+ use the SupCon loss in training. We use the same network structure and embedding dimension, while varying the training objective.

Table 1: OOD detection performance for CIFAR-100 (ID) with ResNet-34. Training with `CIDER` significantly improves OOD detection performance.

| Method | OOD Dataset | | | | | | | | | | Average | |
| | SVHN | | Places365 | | LSUN | | iSUN | | Texture | | | |
| | FPR↓ | AUROC↑ | FPR↓ | AUROC↑ | FPR↓ | AUROC↑ | FPR↓ | AUROC↑ | FPR↓ | AUROC↑ | FPR↓ | AUROC↑ |
|---|---|---|---|---|---|---|---|---|---|---|---|---|
| **Without Contrastive Learning** | | | | | | | | | | | | |
| MSP | 78.89 | 79.80 | 84.38 | 74.21 | 83.47 | 75.28 | 84.61 | 74.51 | 86.51 | 72.53 | 83.12 | 75.27 |
| ODIN | 70.16 | 84.88 | 82.16 | 75.19 | 76.36 | 80.10 | 79.54 | 79.16 | 85.28 | 75.23 | 78.70 | 79.11 |
| Mahalanobis | 87.09 | 80.62 | 84.63 | 73.89 | 84.15 | 79.43 | 83.18 | 78.83 | 61.72 | 84.87 | 80.15 | 79.53 |
| Energy | 66.91 | 85.25 | 81.41 | 76.37 | 59.77 | 86.69 | 66.52 | 84.49 | 79.01 | 79.96 | 70.72 | 82.55 |
| GODIN | 74.64 | 84.03 | 89.13 | 68.96 | 93.33 | 67.22 | 94.25 | 65.26 | 86.52 | 69.39 | 87.57 | 70.97 |
| LogitNorm | 59.60 | 90.74 | 80.25 | 78.58 | 81.07 | 82.99 | 84.19 | 80.77 | 86.64 | 75.60 | 78.35 | 81.74 |
| **With Contrastive Learning** | | | | | | | | | | | | |
| ProxyAnchor | 87.21 | 82.43 | 70.10 | 79.84 | 37.19 | 91.68 | 70.01 | 84.96 | 65.64 | 84.99 | 66.03 | 84.78 |
| CE + SimCLR | 24.82 | 94.45 | 86.63 | 71.48 | 56.40 | 89.00 | 66.52 | 83.82 | 63.74 | 82.01 | 59.62 | 84.15 |
| CSI | 44.53 | 92.65 | 79.08 | 76.27 | 75.58 | 83.78 | 76.62 | 84.98 | 61.61 | 86.47 | 67.48 | 84.83 |
| SSD+ | 31.19 | 94.19 | 77.74 | 79.90 | 79.39 | 85.18 | 80.85 | 84.08 | 66.63 | 86.18 | 67.16 | 85.90 |
| KNN+ | 39.23 | 92.78 | 80.74 | 77.58 | 48.99 | 89.30 | 74.99 | 82.69 | 57.15 | 88.35 | 60.22 | 86.14 |
| CIDER | 23.09 | 95.16 | 79.63 | 73.43 | 16.16 | 96.33 | 71.68 | 82.98 | 43.87 | 90.42 | **46.89** | **87.67** |

As shown in Table 1, OOD detection performance is significantly improved with `CIDER`. Three trends can be observed: (1) Compared to `SSD+` and `KNN+`, `CIDER` explicitly optimizes for inter-class dispersion and leads more desirable embeddings. Moreover, `CIDER` alleviates the class-conditional Gaussian assumptions for OOD detection. Instead, a simple non-parametric distance-based score suffices. Specifically, `CIDER` outperforms the competitive methods `SSD+` by **20.3%** and `KNN+` by **13.3%** in FPR95; (2) While `CSI` (Tack et al., 2020) relies on sophisticated data augmentations and ensembles in testing, `CIDER` only uses the default data augmentations and thus is simpler in practice. Performance wise, `CIDER` reduces the average FPR95 by $20.6\%$ compared to `CSI`; (3) Lastly, as a result of the improved embedding quality, `CIDER` improves the ID accuracy by $0.76\%$ compared to training with the `CE` loss (Table 9). We provide results on the less challenging task (CIFAR-10 as ID) in Appendix E, where CIDER's strong performance remains.

**`CIDER` benefits different distance-based scores.** We show that `CIDER`'s strong representation can benefit different distance-based OOD scores. We consider the Mahalanobis score (denoted as Maha) due to its commonality, and to ensure a fair comparison with `SSD+` under the same OOD score. The results are shown in Table 2. Under *both* KNN (non-parametric) and Maha (parametric) scores, `CIDER` consistently improves the OOD detection performance compared to training with SupCon. For example, `CIDER` with KNN significant reduces FPR95 by $13.33\%$ compared to SupCon+KNN. Moreover, compared to SSD+ (SupCon+Maha), CIDER+Maha reduces FPR95 by **22.77%**. This further highlights the improved representation quality and generality of `CIDER`.

Table 2: Ablation on OOD detection score. Results are FPR95 on CIFAR-100 (ID) with ResNet-34. We evaluate both Mahalanobis and KNN score ($K = 300$).

| Method | OOD Dataset | | | | | AVG FPR95 |
| | SVHN | Places365 | LSUN | iSUN | Texture | |
|---|---|---|---|---|---|---|
| SupCon+Maha (SSD+) | 31.19 | 77.74 | 79.39 | 80.85 | 66.63 | 67.16 |
| CIDER+Maha | **16.68** | **80.34** | **11.07** | **73.82** | **40.06** | **44.39** |
| SupCon+KNN (KNN+) | 39.23 | 80.74 | 48.99 | 74.99 | 57.15 | 60.22 |
| CIDER+KNN | **23.09** | **79.63** | **16.16** | **71.68** | **43.87** | **46.89** |

**Inter-class dispersion is key to strong OOD detection.** Here we examine the effects of loss components on OOD detection. As shown in Table 3, we have the following observations: (1) For ID classification, training with $\mathcal{L}_{\text{comp}}$ alone leads to an accuracy of $75.19\%$, similar to the ID accuracy of `SSD+` ($75.11\%$). This suggests that promoting intra-class compactness and a moderate level of inter-class dispersion (as a result of sample-to-prototype negative pairs in $\mathcal{L}_{\text{comp}}$) are sufficient to discriminate different ID classes; (2) For OOD detection, further inter-class dispersion is critical, which is explicitly encouraged through the dispersion loss $\mathcal{L}_{\text{dis}}$. As a result, adding $\mathcal{L}_{\text{dis}}$ improved the average AUROC by $2\%$. However, promoting inter-class dispersion via $\mathcal{L}_{\text{dis}}$ alone without $\mathcal{L}_{\text{comp}}$ is not sufficient for neither ID classification nor OOD detection. Our ablation suggests that $\mathcal{L}_{\text{dis}}$ and $\mathcal{L}_{\text{comp}}$ work synergistically to improve the hyperspherical embeddings that are desirable for both ID classification and OOD detection.

Table 3: Ablation study on loss component. Results (in AUROC) are based on CIFAR-100 trained with ResNet-34. Training with only $\mathcal{L}_{\text{comp}}$ suffices for ID classification. Inter-class dispersion induced by $\mathcal{L}_{\text{dis}}$ is key to OOD detection.

| Loss Components | | AUROC↑ | | | | | | ID ACC↑ |
|---|---|---|---|---|---|---|---|---|
| $\mathcal{L}_{\text{comp}}$ | $\mathcal{L}_{\text{dis}}$ | Places365 | LSUN | iSUN | Texture | SVHN | AVG | |
| ✓ | | 79.63 | 85.75 | 84.45 | 87.21 | 91.33 | 85.67 | 75.19 |
| | ✓ | 54.76 | 69.81 | 54.99 | 44.26 | 46.48 | 54.06 | 2.03 |
| ✓ | ✓ | **73.43** | **96.33** | **82.98** | **90.42** | **95.16** | **87.67** | **75.35** |

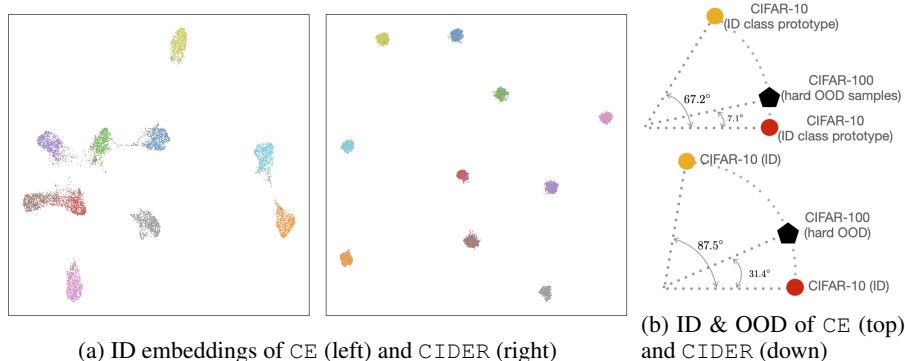

(a) ID embeddings of `CE` (left) and `CIDER` (right)

(b) ID & OOD of `CE` (top) and `CIDER` (down)

Figure 3: (a): UMAP (McInnes et al., 2018) visualization of the features when the model is trained with `CE` vs. `CIDER` for CIFAR-10 (ID). (b): `CIDER` makes OOD samples more separable from ID compared to `CE` (*c.f.* Table 4).

## 4.3 Characterizing and Understanding Embedding Quality

**CIDER learns distinguishable representations.** We visualize the learned feature embeddings in Figure 3 using UMAP (McInnes et al., 2018), where the colors encode different class labels. A salient observation is that embeddings obtained with `CIDER` enjoy much better compactness compared to embeddings trained with the `CE` loss (3a). Moreover, the classes are distributed more uniformly in the space, highlighting the efficacy of the dispersion loss.

**CIDER improves inter-class dispersion and intra-class compactness.** Beyond visualization, we also quantitatively measure the embedding quality. We propose two measurements: inter-class dispersion and intra-class compactness:

$$\text{Dispersion}(\boldsymbol{\mu}) = \frac{1}{C} \sum_{i=1}^{C} \frac{1}{C-1} \sum_{j=1}^{C} \boldsymbol{\mu}_i^\top \boldsymbol{\mu}_j \mathbb{1}\{j \neq i\}.$$

$$\text{Compactness}(\mathcal{D}_{\text{tr}}^{\text{in}}, \boldsymbol{\mu}) = \frac{1}{C} \sum_{j=1}^{C} \frac{1}{n} \sum_{i=1}^{n} \mathbf{z}_i^\top \boldsymbol{\mu}_j \mathbb{1}\{y_i = j\},$$

where $\mathcal{D}_{\text{tr}}^{\text{in}} = \{\mathbf{x}_i, y_i\}_{i=1}^{N}$, and $\mathbf{z}_i$ is the normalized embedding of $\mathbf{x}_i$ for all $1 \leq i \leq N$. Dispersion is measured by the average cosine similarity among pair-wise class prototypes. The compactness can be interpreted as the average cosine similarity between each feature embedding and its corresponding class prototype.

To make the measurements more interpretable, we convert cosine similarities to *angular degrees*. Hence, a higher inter-class dispersion (in degrees) indicates more separability among class prototypes, which is desirable. Similarly, lower intra-class compactness (in degrees) is better. The results are shown in Table 4 based on the CIFAR-10 test set. Compared to `SSD+` (with `SupCon` loss), `CIDER` significantly improves the inter-class dispersion by **12.03** degrees. Different from `SupCon`, `CIDER` explicitly optimizes the inter-class dispersion, which especially benefits OOD detection.

**CIDER improves ID-OOD separability.** Next, we quantitatively measure how the feature embedding quality affects the ID-OOD separability. We introduce a *separability score*, which measures

Table 4: Compactness and dispersion of CIFAR-10 feature embedding, along with the separability *w.r.t.* each OOD test set. We convert cosine similarity to angular degrees for better readability.

| Training Loss | Dispersion (ID)↑ | Compactness (ID)↓ | ID-OOD Separability↑ (in degree) | | | | | |
|---|---|---|---|---|---|---|---|---|
| | (in degree) | (in degree) | CIFAR-100 | LSUN | iSUN | Texture | SVHN | **AVG** |
| Cross-Entropy | 67.17 | 24.53 | 7.11 | 14.57 | 13.70 | 13.76 | 11.08 | 12.04 |
| SSD+ (SupCon loss) | 75.50 | 22.08 | 23.90 | 28.55 | 25.70 | 33.45 | 37.70 | 29.86 |
| CIDER (ours) | **87.53** | **21.35** | **31.41** | **48.37** | **41.54** | **39.60** | **51.65** | **42.51** |

on average how close the embedding of a sample from the OOD test set is to the closest ID class prototype, compared to that of an ID sample. The traditional notion of "OOD being far away from ID classes" is now translated to "OOD being somewhere between ID clusters on the hypersphere". A higher separability score indicates that the OOD test set is easier to be detected. Formally, we define the separability measurement as:

$$\uparrow \text{Separability} = \frac{1}{|\mathcal{D}_{\text{test}}^{\text{ood}}|} \sum_{\mathbf{x} \in \mathcal{D}_{\text{test}}^{\text{ood}}} \max_{j \in [C]} \mathbf{z}_{\mathbf{x}}^{\top} \boldsymbol{\mu}_j - \frac{1}{|\mathcal{D}_{\text{test}}^{\text{in}}|} \sum_{\mathbf{x}' \in \mathcal{D}_{\text{test}}^{\text{in}}} \max_{j \in [C]} \mathbf{z}_{\mathbf{x}'}^{\top} \boldsymbol{\mu}_j, \quad (9)$$

where $\mathcal{D}_{\text{test}}^{\text{ood}}$ is the OOD test dataset and $\mathbf{z}_{\mathbf{x}}$ denotes the normalized embedding of sample $\mathbf{x}$. Table 4 shows that CIDER leads to higher separability and consequently superior OOD detection performance (*c.f.* Table 1). Averaging across 5 OOD test datasets, our method displays a relative **42.36%** improvement of ID-OOD separability compared to `SupCon`. This further verifies the effectiveness of `CIDER` for improving OOD detection.

## 4.4 ADDITIONAL ABLATIONS AND ANALYSIS

**CIDER is competitive on large-scale datasets.** To further examine the performance of CIDER on real-world tasks, we evaluate the performance of CIDER on the more challenging large-scale benchmarks. Specifically, we use ImageNet-100 as ID, a subset of ImageNet (Deng et al., 2009) consisting of 100 randomly sampled classes. For OOD test datasets, we use the same ones in (Huang & Li, 2021), including subsets of iNATURALIST (Van Horn et al., 2018), SUN (Xiao et al., 2010), PLACES (Zhou et al., 2017), and TEXTURE (Cimpoi et al., 2014a). For each OOD dataset, the categories do not overlap with the ID dataset. For computational efficiency, we fine-tune pretrained ResNet-34 with CIDER and SupCon losses for 10

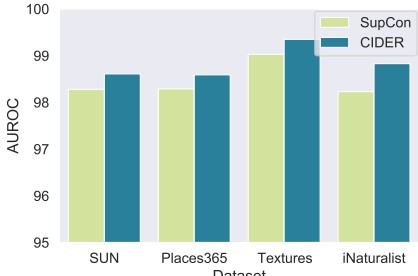

Figure 4: Fine-tuning pre-trained ResNet-34 on ImageNet-100 (ID).

epochs with an initial learning rate of 0.01. For each loss, we update the weights of the last residual block and the nonlinear projection head, while freezing the parameters in the first three residual blocks. At test time, we use the same detection score (KNN) to evaluate representation quality. The performance (in AUROC) is shown in Figure 4 (more results are in Appendix D). We can see that CIDER remains very competitive on all the OOD test sets where CIDER consistently outperforms SupCon. This further verifies the benefits of explicitly promoting inter-class dispersion and intra-class compactness.

**Ablation studies on weight scale, prototype update factor, learning rate, temperature, batch size, and architecture.** We provide comprehensive ablation studies to understand the impact of various factors in Appendix C. For example, as shown in Figure 5, CIDER consistently outperforms SupCon across different batch sizes.

## 5 RELATED WORKS

**Out-of-distribution detection.** The majority of works in the OOD detection literature focus on the supervised setting,

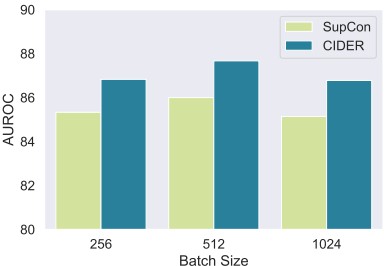

Figure 5: OOD Detection performance across different batch sizes.

where the goal is to derive a binary ID-OOD classifier along with a classification model for the in-distribution data. Compared to generative model-based methods (Kirichenko et al., 2020; Nalisnick et al., 2019; Ren et al., 2019; Serrà et al., 2020; Xiao et al., 2020), OOD detection based on supervised discriminative models typically yield more competitive performance. For deep neural networks-based methods, most OOD detection methods derive confidence scores either based on the output (Bendale & Boult, 2016; Hendrycks & Gimpel, 2017; Hsu et al., 2020; Huang & Li, 2021; Liang et al., 2018; Liu et al., 2020; Sun et al., 2021; Ming et al., 2022b), gradient information (Huang et al., 2021), or the feature embeddings (Lee et al., 2018; Sastry & Oore, 2020; Tack et al., 2020; Zhou et al., 2021; Sehwag et al., 2021; Sun et al., 2022; Ming et al., 2022a; Du et al., 2022). Our method can be categorized as a distance-based OOD detection method by exploiting the hyperspherical embedding space.

**Contrastive representation learning.** Contrastive representation learning (van den Oord et al., 2019) aims to learn a discriminative embedding where positive samples are aligned while negative ones are dispersed. It has demonstrated remarkable success for visual representation learning in unsupervised (Chen et al., 2020a;b; He et al., 2020; Robinson et al., 2021), semi-supervised (Assran et al., 2021), and supervised settings (Khosla et al., 2020). Recently, Li et al. (2021b) propose a prototype-based contrastive loss for unsupervised learning where prototypes are generated via a clustering algorithm, while our method is supervised where prototypes are updated based on labels. Li et al. (2021a) incorporate a prototype-based loss to tackle data noise. Wang & Isola (2020) analyze the asymptotic behavior of contrastive losses theoretically, while Wang & Liu (2021a) empirically investigate the properties of contrastive losses for classification. Recently, Bucci et al. (2022) investigates contrastive learning for open-set domain adaptation. However, none of the works focus on OOD detection. We aim to fill the gap and facilitate the design and understanding of contrastive losses for OOD detection.

**Representation learning for OOD detection.** Self-supervised learning has been shown to improve OOD detection. Prior works (Sehwag et al., 2021; Winkens et al., 2020) verify the effectiveness of directly applying the off-the-shelf multi-view contrastive losses such as `SupCon` and `SimCLR` for OOD detection. `CSI` (Tack et al., 2020) investigates the type of data augmentations that are particularly beneficial for OOD detection. Different from prior works, we focus on hyperspherical embeddings and propose to explicitly encourage the desirable properties for OOD detection, and thus alleviate the dependence on specific data augmentations or self-supervision. Moreover, CIDER explicitly models the latent representations as vMF distributions, providing a direct and clear geometrical interpretation of hyperspherical embeddings. Closest to our work, Du et al. (2022) recently explores shaping representations into vMF distributions for object-level OOD detection. However, they do not consider the inter-class dispersion loss, which we show is crucial to achieve strong OOD detection performance.

**Deep metric learning.** Learning a desirable embedding is a fundamental goal in the deep metric learning community. Various losses have been proposed for face verification (Deng et al., 2019; Liu et al., 2017; Wang et al., 2018), person re-identification (Chen et al., 2017; Xiao et al., 2017), and image retrieval (Kim et al., 2019; Movshovitz-Attias et al., 2017; Oh Song et al., 2016; Teh et al., 2020). However, none of the works focus on desirable embeddings for OOD detection. The difference between CIDER and proxy-based methods has been discussed in Remark 2.

## 6 CONCLUSION AND OUTLOOK

In this work, we propose `CIDER`, a novel representation learning framework that exploits hyperspherical embeddings for OOD detection. `CIDER` jointly optimizes the *dispersion* and *compactness* losses to promote strong ID-OOD separability. We show that `CIDER` achieves superior performance on common OOD benchmarks, including large-scale OOD detection tasks. Moreover, we introduce new measurements to quantify the hyperspherical embedding, and establish the relationship with OOD detection performance. We conduct extensive ablations to understand the efficacy and behavior of `CIDER` under various settings and hyperparameters. We hope our work can inspire future methods of exploiting hyperspherical representations for OOD detection.

ACKNOWLEDGEMENT

We gratefully acknowledge the support of AFOSR Young Investigator Award under No. FA9550-23-1-0184; Philanthropic Fund from SFF; and faculty research awards from Google, Meta, and Amazon. Any opinions, findings, conclusions, or recommendations expressed in this material are those of the authors and do not necessarily reflect the views, policies, or endorsements either expressed or implied, of the sponsors.

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

APPENDIX

## A  ALGORITHM DETAILS AND DISCUSSIONS

The training scheme of our compactness and dispersion regularized (CIDER) learning framework is shown in Algorithm 1. We jointly optimize: (1) a *compactness loss* to encourage samples to be close to their class prototypes, and (2) a *dispersion loss* to encourage larger angular distances among different class prototypes.

---

**Algorithm 1:** Pseudo-code of CIDER.

1 **Input:** Training dataset $\mathcal{D}$, neural network encoder $f$, projection head $h$, classifier $g$, class prototypes $\boldsymbol{\mu}_j$ ($1 \leq j \leq C$), weights of loss terms $\lambda_d$, and $\lambda_c$, temperature $\tau$
2 **for** $epoch = 1, 2, \ldots,$ **do**
3     **for** $iter = 1, 2, \ldots,$ **do**
4         sample a mini-batch $B = \{\mathbf{x}_i, y_i\}_{i=1}^b$
5         obtain augmented batch $\tilde{B} = \{\tilde{\mathbf{x}}_i, \tilde{y}_i\}_{i=1}^{2b}$ by applying two random augmentations to $\mathbf{x}_i \in B$
            $\forall i \in \{1, 2, \ldots, b\}$
6         **for** $\tilde{\mathbf{x}}_i \in \tilde{B}$ **do**
            // obtain normalized embedding
7             $\tilde{\mathbf{z}}_i = h(f(\tilde{\mathbf{x}}_i)), \mathbf{z}_i = \tilde{\mathbf{z}}_i / \|\tilde{\mathbf{z}}_i\|_2$
            // update class-prototypes
8             $\boldsymbol{\mu}_c := \text{Normalize}(\alpha \boldsymbol{\mu}_c + (1 - \alpha)\mathbf{z}_i), \, \forall c \in \{1, 2, \ldots, C\}$
        // calculate compactness loss
9         $\mathcal{L}_{\text{comp}} = -\sum_{i=1}^b \log \frac{\exp(\mathbf{z}_i^\top \boldsymbol{\mu}_{c(i)} / \tau)}{\sum_{j=1}^C \exp(\mathbf{z}_i^\top \boldsymbol{\mu}_j / \tau)}$
        // calculate dispersion loss
10        $\mathcal{L}_{\text{dis}} = \frac{1}{C} \sum_{i=1}^C \log \frac{1}{C-1} \sum_{j=1}^C \mathbb{1}\{j \neq i\} e^{\boldsymbol{\mu}_i^\top \boldsymbol{\mu}_j / \tau}$
        // calculate overall loss
11        $\mathcal{L} = \mathcal{L}_{\text{dis}} + \lambda_c \mathcal{L}_{\text{comp}}$
        // update the network weights
12        update the weights in the encoder and the projection head

---

**Remark 1: The prototype update rule.** The class prototypes are only updated by exponential moving average (EMA). Since the prototypes are not learnable parameters, the gradients of the dispersion loss have no direct impact on their updates. EMA-style techniques have been used in prior works Li et al. (2020), and can be rigorously interpreted from a clustering-based Expectation-Maximization (EM) perspective. Alternatively, the prototypes can also be updated via gradients without EMA. We provide an ablation study of CIDER based on different prototype update rules in Appendix C.

**Remark 2: CIDER vs. Wang & Isola (2020).** The notion of alignment and uniformity for contrastive losses were proposed in Wang & Isola (2020) for the *unsupervised setting* where both metrics are based on individual samples. CIDER is a contrastive loss designed for the *supervised* setting. In particular, the uniform loss in Wang & Isola (2020) is defined based on randomly sampled pairs of data and promotes an *instance-to-instance* uniformity on the hypersphere. The notion of uniformity is fundamentally different from CIDER, which promotes *prototype-to-prototype* dispersion.

**Remark 3: CIDER vs. Cross Entropy.** When a model is trained with the cross-entropy (CE) loss, the weight matrix of the last fully connected layer can be interpreted as the set of class prototypes. However, CE is suboptimal for OOD detection for two main reasons: (1) CE does not explicitly optimize for the intra-class compactness and inter-class dispersion in the feature space. As a consequence, the embeddings obtained by CE loss display insufficient compactness and dispersion (Table 4). Compared to CE, CIDER is more structured by exploiting the hyperspherical embeddings and explicitly optimizing towards the desirable properties for OOD detection and ID classification. (2) the feature space obtained by CE loss is Euclidean instead of hyperspherical. As shown in Tack et al. (2020), ID data tend to have a larger norm than OOD data. As a result, the Euclidean distance between ID features can be larger than the distance from OOD to ID data. A recent work (Sun et al., 2022) verified that Euclidean embedding without feature normalization leads

to suboptimal OOD detection performance. Instead, CIDER is designed to optimize hyperspherical embeddings, which benefit OOD detection.

**Remark 4: On the measurement of embedding quality.** In Section 4.3, we provide measurements of embedding quality (inter-class dispersion and intra-class compactness) via prototypes. In practice, one can replace the class prototypes in the dispersion and compactness metrics to be one random sample (or the average of a random subset) from the corresponding class. For example, on CIFAR-10, as the embeddings of CIDER are compact, we observe that the two metrics give similar results (e.g., the ID-OOD Separability is 42.5 with prototype-based metrics vs. 38.3 with instance-based metrics). We choose to use prototypes because (1) the definition directly maps to our loss function design, and (2) prototypes are calculated as the averaged feature for each class, which helps to mitigate the sampling bias.

# B    EXPERIMENTAL DETAILS

**Software and hardware.** All methods are implemented in Pytorch 1.10. We run all the experiments on NVIDIA GeForce RTX-2080Ti GPU for small to medium batch size and on NVIDIA A100 GPU for large batch size and larger network encoder.

**Architecture.** As shown in Figure 1, the overall architecture of `CIDER` consists of a projection head $h$ on top of a deep neural network encoder $f$. Following common practice and fair comparison with prior works (Sehwag et al., 2021), (Khosla et al., 2020), we fix the output dimension of the projection head to be 128. We use a two-layer non-linear projection head for CIFAR-10 and CIFAR-100 as in Sun et al. (2022).

**Training.** For methods based on pre-trained models such as MSP (Hendrycks & Gimpel, 2017), ODIN (Liang et al., 2018), Mahalanobis (Lee et al., 2018), and Energy (Liu et al., 2020), we follow the configurations in Sun et al. (2022) for CIFAR-10 and train with the cross-entropy loss for 100 epochs. The initial learning rate is 0.1 and decays by a factor of 10 at epochs 50, 75, and 90 respectively. For the more challenging dataset CIFAR-100, we train 200 epochs. We use stochastic gradient descent with momentum 0.9, and weight decay $10^{-4}$. For fair comparison, methods involving contrastive learning (Winkens et al., 2020),(Tack et al., 2020),(Sehwag et al., 2021) are trained for 500 epochs on CIFAR-10 and CIFAR-100. For `CIDER`, we adopt the same key hyperparameters for contrastive losses such as initial learning rate (0.5), temperature (0.1), and batch size (512) as `SSD+` (Sehwag et al., 2021) in main experiments to demonstrate the effectiveness and simplicity of CIDER. For the prototype update factor $\alpha$, we set the default value as 0.95 for simplicity. We observed that $\alpha = 0.95$ on CIFAR-10 with ResNet-18 and $\alpha = 0.5$ on CIFAR-100 with ResNet-34 provide stronger performance.

**OOD detection score.** By default, we use the non-parametric KNN score Sun et al. (2022). We use a larger $K = 300$ for CIFAR-100 and a smaller $K = 100$ for CIFAR-10 for simplicity. Adjusting $K \in \{10, 20, 50, 100, 200, 300, 500\}$ yields similar performance. In practice, $K$ can be tuned using a validation method (Sun et al., 2022) to further improve the performance.

# C    ADDITIONAL ABLATION STUDIES

**CIDER is effective under various batch sizes.** Figure 6a and 6b also indicate that `CIDER` remains competitive under different batch size configurations compared to `SupCon`. To explain this, the standard `SupCon` loss requires *instance-to-instance* distance measurement, whereas compactness loss reduces the complexity to *instance-to-prototype*. The class-conditional prototypes are updated during training, which capture the average statistics of each class and alleviate the dependency on the batch size. This leads to an overall memory-efficient solution for OOD detection.

**Ablation on the loss weights.** In the main results (Table 1), we demonstrate the effectiveness of `CIDER` where the loss weight $\lambda_c$ is simply set to balance the initial scale between the $\mathcal{L}_{\text{dis}}$ and $\mathcal{L}_{\text{comp}}$. In fact, `CIDER` can be further improved by adjusting $\lambda_c$. As shown in Figure 7a, the

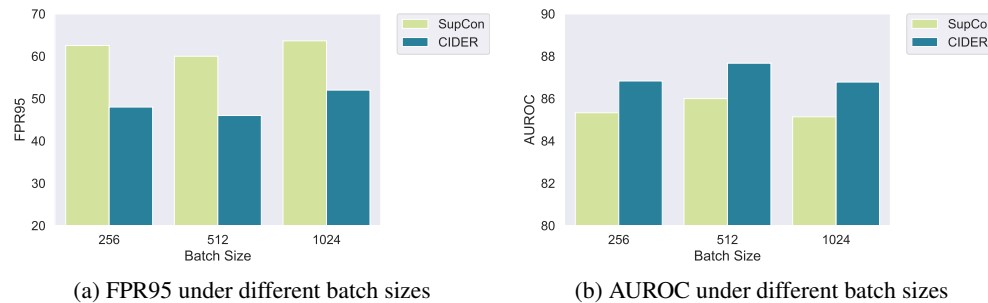

(a) FPR95 under different batch sizes      (b) AUROC under different batch sizes

Figure 6: Ablation on `CIDER` v.s. `SupCon` loss under different batch sizes. The results are averaged across the 5 OOD test sets based on ResNet-34. `CIDER` outperforms `SupCon` across different batch sizes, suggesting the effectiveness of explicitly facilitating prototype-wise dispersion.

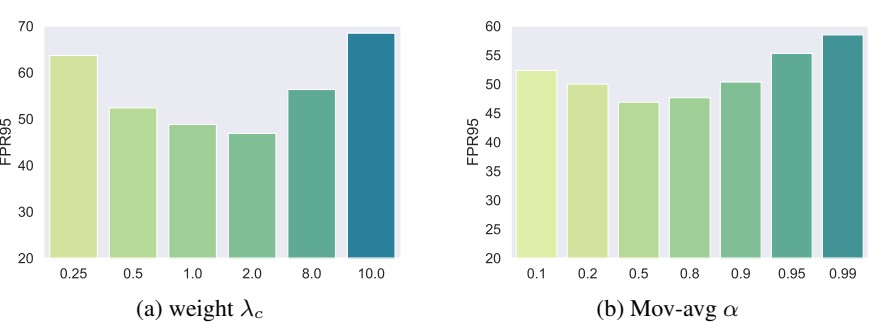

(a) weight $\lambda_c$      (b) Mov-avg $\alpha$

Figure 7: Ablation on (a) weight $\lambda_c$ of the compactness loss; (b) prototype update discount factor $\alpha$. The results are based on CIFAR-100 (ID) averaged over 5 OOD test sets.

performance of `CIDER` is relatively stable for moderate adjustments of $\lambda_c$ (*e.g.* 0.5 to 2), with the best performance at around $\lambda_c \in [1, 2]$. This indicates `CIDER` provides a simple and effective solution for improving OOD detection, without much need for hyperparameter tuning on the loss scale.

**Adjusting prototype update factor $\alpha$ improves CIDER.** We show in Figure 7b the performance by varying the moving-average discount factor $\alpha$ in Eq. 8. We can observe that the detection performance (averaged over 5 test sets) is still competitive across a wide range of $\alpha$. In particular, for CIFAR-100, $\alpha = 0.5$ results in the best performance with average FPR95 of $46.89\%$ under KNN score. For CIFAR-10, we observe that a larger $\alpha$ (*e.g.* 0.95 to 0.99) results in stronger performance.

**Ablation on the learning rate.** Prior works (Khosla et al., 2020; Sehwag et al., 2021) use a default initial learning rate (lr) of 0.5 to train contrastive losses, which is also the default setting of `CIDER`. We further investigate the impact of the initial learning rate on OOD detection. As shown in Figure 8a, a relatively higher initial lr is indeed desirable for competitive performance while too small lr (*e.g.* 0.1) would lead to performance degradation.

**Small temperature $\tau$ leads to better performance.** Figure 8b demonstrates the detection performance as we vary the temperature parameter $\tau$. We observe that the OOD detection performance is desirable at a relatively small temperature. Complementary to our finding, a relatively small temperature is shown to be desirable for ID classification (Khosla et al., 2020; Wang & Liu, 2021b) which penalizes hard negative samples with larger gradients and leads to separable features.

**Ablation on network capacity.** We verify the effectiveness of `CIDER` under networks with other architectures such as ResNet-50 for CIFAR-100. The results are shown in Figure 9. The trend is similar to what we observed with ResNet-34. Specifically, as a result of the improved representation, training with `CIDER` improves the FPR95 for various test sets compared to training with the `SupCon` loss.

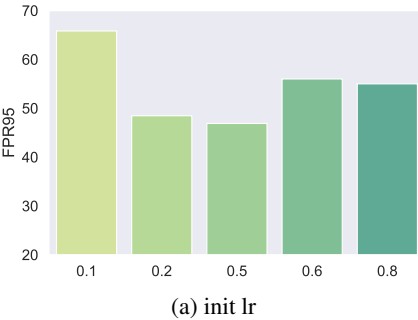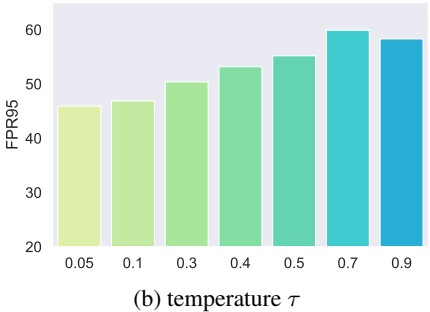

Figure 8: Ablation on (a) initial learning rate and (b) temperature. The results are based on CIFAR-100 (ID) averaged over 5 OOD test sets.

**Ablation on gradient-based prototype update.** We examine the effect of updating prototypes via gradients. Compared to CIDER with EMA, CIDER with learnable prototypes (LP) can be more sensitive to initialization. We report the average performance of CIDER (EMA) and CIDER (LP) across 3 independent runs for CIFAR-10 in Table 5. All training and evaluation configurations (e.g., learning rate and batch size) are the same. We can see that CIDER with EMA improves the average FPR95 by $5.08\%$ with smaller standard deviation. Therefore, we empirically verify that updating prototypes via EMA is a better option with stronger training stability in practice.

Table 5: Ablation on prototype update rules. OOD detection performance for ResNet-18 trained on CIFAR-10 with EMA-style updates denoted as CIDER (EMA) vs. learnable prototypes denoted as CIDER (LP). CIDER with EMA demonstrates strong OOD detection performance. Results are averaged over 3 independent runs.

| Method | SVHN | | Places365 | | OOD Dataset
LSUN | | iSUN | | Texture | | Average | |
|---|---|---|---|---|---|---|---|---|---|---|---|---|
| | FPR↓ | AUROC↑ | FPR↓ | AUROC↑ | FPR↓ | AUROC↑ | FPR↓ | AUROC↑ | FPR↓ | AUROC↑ | FPR↓ | AUROC↑ |
| CIDER (LP) | $2.17^{\pm1.50}$ | $99.55^{\pm0.48}$ | $28.13^{\pm1.59}$ | $94.53^{\pm0.12}$ | $5.23^{\pm2.68}$ | $98.16^{\pm0.99}$ | $36.47^{\pm8.93}$ | $94.59^{\pm0.92}$ | $16.25^{\pm1.07}$ | $97.38^{\pm0.19}$ | $17.65^{\pm2.36}$ | $96.84^{\pm0.40}$ |
| CIDER (EMA) | $3.04^{\pm1.38}$ | $99.50^{\pm0.30}$ | $26.60^{\pm2.47}$ | $94.64^{\pm0.51}$ | $4.10^{\pm1.68}$ | $99.14^{\pm0.19}$ | $15.94^{\pm4.56}$ | $97.10^{\pm0.54}$ | $13.19^{\pm0.82}$ | $97.39^{\pm0.48}$ | $12.57^{\pm1.31}$ | $97.56^{\pm0.33}$ |

**Stability of CIDER.** To verify that CIDER consistently provides strong performance, we train with 3 independent seeds for each ID dataset. Table 6 shows the OOD detection performance of CIDER with ResNet-18 trained on CIFAR-10 and ResNet-34 trained on CIFAR-100. Comparing Table 1 and Table 7, we can see that CIDER yields consistently strong performance. Code and checkpoints are provided in `https://github.com/deeplearning-wisc/cider`.

Table 6: Ablation on stability. OOD detection performance of CIDER for CIFAR-10 and CIFAR-100. Results are averaged over 3 independent runs.

| ID Dataset | SVHN | | Places365 | | OOD Dataset
LSUN | | iSUN | | Texture | | Average | |
|---|---|---|---|---|---|---|---|---|---|---|---|---|
| | FPR↓ | AUROC↑ | FPR↓ | AUROC↑ | FPR↓ | AUROC↑ | FPR↓ | AUROC↑ | FPR↓ | AUROC↑ | FPR↓ | AUROC↑ |
| CIFAR-10 | $3.04^{\pm1.38}$ | $99.50^{\pm0.30}$ | $26.60^{\pm2.47}$ | $94.64^{\pm0.51}$ | $4.10^{\pm1.68}$ | $99.14^{\pm0.19}$ | $15.94^{\pm4.56}$ | $97.10^{\pm0.54}$ | $13.19^{\pm0.82}$ | $97.39^{\pm0.48}$ | $12.57^{\pm1.31}$ | $97.56^{\pm0.33}$ |
| CIFAR-100 | $23.67^{\pm2.28}$ | $95.07^{\pm0.13}$ | $79.37^{\pm1.84}$ | $72.97^{\pm3.90}$ | $22.04^{\pm5.12}$ | $96.01^{\pm1.80}$ | $62.16^{\pm8.48}$ | $83.70^{\pm2.92}$ | $44.96^{\pm6.01}$ | $90.25^{\pm0.97}$ | $46.45^{\pm2.01}$ | $87.60^{\pm1.03}$ |

# D RESULTS ON LARGE-SCALE DATASETS

In recent years, there has been a paradigm shift towards fine-tuning pre-trained models, as opposed to training from scratch. Given this trend, it is important to explore whether CIDER remains effective based on pre-trained models. Specifically, we fine-tune ImageNet pre-trained ResNet-34 on ImageNet-100 with CIDER and SupCon losses for 10 epochs. For each loss, we update the weights of the last residual block and the nonlinear projection head, while freezing the parameters in the first three residual blocks. At test time, we use the same detection score (KNN) to evaluate representation quality. FPR95 and AUROC for each OOD test set are shown in Figure 10a and 10b, respectively. The results suggest that CIDER remains very competitive, which highlight the benefits of promoting inter-class dispersion and intra-class compactness.

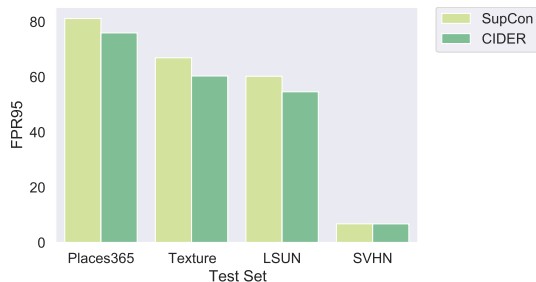

Figure 9: Ablation on architecture. Results are based on ResNet-50.

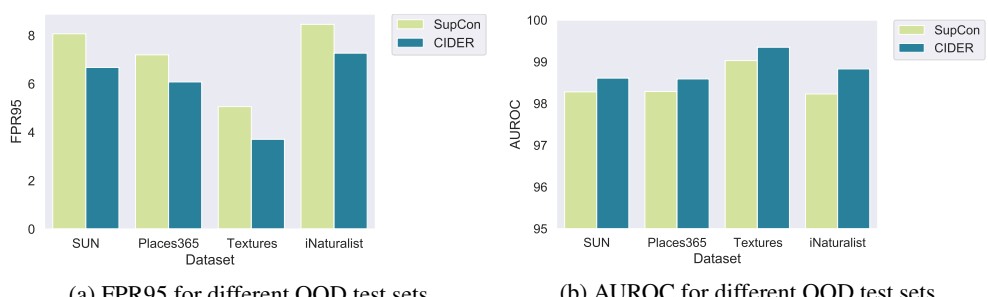

(a) FPR95 for different OOD test sets

(b) AUROC for different OOD test sets

Figure 10: OOD detection performance of fine-tuning with CIDER v.s. SupCon for ImageNet-100 (ID). With the same detection score (KNN), CIDER consistently outperforms SupCon across all OOD test datasets.

# E    RESULTS ON CIFAR-10

In the main paper, we mainly focus on the more challenging task CIFAR-100. In this section, we additionally evaluate on CIFAR-10, a commonly used benchmark in literature. For methods involving contrastive losses, we use the same network encoder and embedding dimension, while only varying the training objective. The Mahalanobis score is used for OOD detection in `SSD+` (Sehwag et al., 2021), `CE+SimCLR` (Winkens et al., 2020), `SupCon`, and `CIDER`. As CIFAR-10 is much less challenging compared to CIFAR-100, recent methods with contrastive losses yield similarly strong performance. For methods trained with cross-entropy loss, we use the publicly available checkpoints in Sun et al. (2022) for better consistency. The results are shown in Table 7. Similar trends also hold as we describe in Section 4.2: (1) `CIDER` achieves superior OOD detection performance in CIFAR-10 as a result of better inter-class dispersion and intra-class compactness. For example, compared to the Mahalanobis baseline (Lee et al., 2018), `CIDER` reduces the FPR95 by $24.99\%$ averaged over 5 diverse test sets; (2) Although the ID classification accuracy of `CIDER` is similar to another proxy-based loss `ProxyAnchor` (Table 8), `CIDER` significantly improves the OOD detection performance by $21.05\%$ in FPR95 due to the addition of explicit inter-class dispersion which we show is critical for OOD detection in Section 4.3. The significant improvements highlight the importance of representation learning for OOD detection.

# F    ID CLASSIFICATION ACCURACY

The ID classification accuracy on CIFAR-10 and CIFAR-100 can be seen in Table 8 and Table 9, where for contrastive losses such as KNN+, SSD+, and CIDER, we follow the common practice as in Khosla et al. (2020) and use linear probe on normalized features.

| Method | OOD Dataset | | | | | | | | | | Average | |
| | SVHN | | Places365 | | LSUN | | iSUN | | Texture | | | |
| | FPR↓ | AUROC↑ | FPR↓ | AUROC↑ | FPR↓ | AUROC↑ | FPR↓ | AUROC↑ | FPR↓ | AUROC↑ | FPR↓ | AUROC↑ |
|---|---|---|---|---|---|---|---|---|---|---|---|---|
| **Without Contrastive Learning** | | | | | | | | | | | | |
| MSP | 59.66 | 91.25 | 62.46 | 88.64 | 45.21 | 93.80 | 54.57 | 92.12 | 66.45 | 88.50 | 57.67 | 90.86 |
| Energy | 54.41 | 91.22 | 42.77 | 91.02 | 10.19 | 98.05 | 27.52 | 95.59 | 55.23 | 89.37 | 38.02 | 93.05 |
| ODIN | 53.78 | 91.30 | 43.40 | 90.98 | 10.93 | 97.93 | 28.44 | 95.51 | 55.59 | 89.47 | 38.43 | 93.04 |
| GODIN | 18.72 | 96.10 | 55.25 | 85.50 | 11.52 | 97.12 | 30.02 | 94.02 | 33.58 | 92.20 | 29.82 | 92.97 |
| Mahalanobis | 9.24 | 97.80 | 83.50 | 69.56 | 67.73 | 73.61 | 6.02 | 98.63 | 23.21 | 92.91 | 37.94 | 86.50 |
| **With Contrastive Learning** | | | | | | | | | | | | |
| CE + SimCLR | 6.98 | 99.22 | 54.39 | 86.70 | 64.53 | 85.60 | 59.62 | 86.78 | 16.77 | 96.56 | 40.46 | 90.97 |
| CSI | 37.38 | 94.69 | 38.31 | 93.04 | 10.63 | 97.93 | 10.36 | 98.01 | 28.85 | 94.87 | 25.11 | 95.71 |
| SSD+ | 2.47 | 99.51 | 22.05 | 95.57 | 10.56 | 97.83 | 28.44 | 95.67 | 9.27 | 98.35 | 14.56 | 97.38 |
| ProxyAnchor | 39.27 | 94.55 | 43.46 | 92.06 | 21.04 | 97.02 | 23.53 | 96.56 | 42.70 | 93.16 | 34.00 | 94.67 |
| KNN+ | 2.70 | 99.61 | 23.05 | 94.88 | 7.89 | 98.01 | 24.56 | 96.21 | 10.11 | 97.43 | 13.66 | 97.22 |
| CIDER | 2.89 | 99.72 | 23.88 | 94.09 | 5.45 | 99.01 | 20.21 | 96.64 | 12.33 | 96.85 | 12.95 | 97.26 |

Table 7: Results on CIFAR-10. OOD detection performance for ResNet-18 trained on CIFAR-10 with and without contrastive loss. `CIDER` achieves strong OOD detection performance and ID classification accuracy (Table 8).

Table 8: ID classification accuracy on CIFAR-10 (%)

| Method | ID ACC |
|---|---|
| **w.o. contrastive loss** | |
| MSP | 94.21 |
| ODIN | 94.21 |
| GODIN | 93.64 |
| Energy | 94.21 |
| Mahalanobis | 94.21 |
| **w. contrastive loss** | |
| CE + SimCLR | 93.12 |
| SSD+ | 94.53 |
| ProxyAnchor | 94.21 |
| KNN+ | 94.53 |
| CIDER | 94.58 |

Table 9: ID classification accuracy on CIFAR-100 (%)

| Method | ID ACC |
|---|---|
| **w.o. contrastive loss** | |
| MSP | 74.59 |
| ODIN | 74.59 |
| GODIN | 74.92 |
| Energy | 74.59 |
| Mahalanobis | 74.59 |
| **w. contrastive loss** | |
| CE + SimCLR | 73.54 |
| SSD+ | 75.11 |
| ProxyAnchor | 74.21 |
| KNN+ | 75.11 |
| CIDER | 75.35 |

