# OpenReview forum: "How to Exploit Hyperspherical Embeddings for Out-of-Distribution Detection?"
_ICLR.cc/2023/Conference — ICLR 2023 poster_

### Official Review · Reviewer_FAUX · 2022-10-22

**Confidence:** 3
**Correctness:** 3
**Technical Novelty And Significance:** 2
**Empirical Novelty And Significance:** 3
**Recommendation:** 6

**Clarity, Quality, Novelty And Reproducibility:**

The paper would benefit from additional details (regarding experiments, see W3, W4) and better explanations and relations to prior work (see W1 and W2).



**Strength And Weaknesses:**

**Strong points**:

S1. The method obtains good results for OOD detection compared to recent methods.

S2. The method and investigations are sound.

S3. Paper is mainly well-written.

**Weak Points**:

W1. The relation between the proposed losses and other common losses (cross-entropy and contrastive losses) should be better explained and investigated more. For example, the goals presented in paragraph Training objective (Sec. 3.2) is also attained by cross entropy loss, why should the proposed method behave better? The last weight matrix of a deep model could be interpreted as the set of class prototypes ( with cross-entropy loss thus being similar to the compactness loss)  what are the advantages of the proposed approach of using moving averages as propotypes. The Alignmentand Uniformity metrics of Wang and Isola 2020 share similar goals as the proposed losses and should be discussed also.

W2. The paper proposes some experimental investigations into the benefits of the method, and they are appreciated, but they should be expanded and better explained.

 W2.1. While it is perfectly sound to optimize the embeddings using the class prototypes, it is not clear why the measurements of embedding quality (inter-class dispersion and intra-class compactness from Sec 4.3) should take the prototypes into account. More fundamental metrics should be defined based on similarity between embeddings of samples $z_i$ and $z_j$ of different classes (for dispersion) or the same class (for Compactness) (similar to Wang and Isola 2020). Similarly, the separability metric could be designed based on average similarities between pairs of embeddings. Using the prototypes is just a proxy of these, more fundamental metrics. So why should be metrics be defined based on prototypes?

 W2.2 How are prototypes defined for Cross-Entropy SSD+ methods shown in Table 4? Are they computed by eq.8 during training? For the Cross-entropy method, are the embeddings $L_2$ normalized during training and when computing the metrics? If they are normalized only for the metrics, does it make sense to use them in this new hyperspherical space if they were not trained in this space?



W3. How are the results in Table 1 /  Table 6 computed? Are the other methods trained with original code or are the original results presented? Table 6 shows worse results for all other methods than a similar table (Table 1) from (Sun et al. 2022). Why?

W4. How many seeds were used when computing the results? It would be better to use multiple seeds and present results with mean and standard deviations.

W5. It should be noted that this method needs ID class labels, having a disadvantage as opposed to contrastive methods that are designed to work both with and without labels.



**Summary Of The Paper:**

The paper proposes a new method (CIDER) for representation learning and applies it for out-of-distribution detection. Given an $L_2$ normalized embedding (hyperspherical embeddings) from a deep network, the method computes prototypes, as moving averages of the embeddings corresponding to each class. The method is optimized using two losses. The first one (compactness loss) minimizes the distance between a sample and the prototype corresponding to its class while maximizing the distance to the other ones. The second loss (dispersionloss loss) maximizes the distance between all pairs of class prototypes. The representations learned by this method are then used to detect out-of-distribution samples using KNN distance.



**Summary Of The Review:**

This paper proposes a sound method to obtain representations useful for OOD detection. Some further explanations and investigations into the benefits of the proposed method are still needed.

---

### Official Review · Reviewer_U6bR · 2022-10-25

**Confidence:** 3
**Correctness:** 3
**Technical Novelty And Significance:** 3
**Empirical Novelty And Significance:** 3
**Recommendation:** 6

**Clarity, Quality, Novelty And Reproducibility:**

This paper is well written and easy to follow. The main idea is presented clearly. Details of experiment settings can be found in the paper,  and corresponding analysis support the claim well.

**Strength And Weaknesses:**

Strength:
1. The idea is well motivated and writing is good
2. Experiment evaluation is thorough

Weakness:
1. Training scheme needs further analysis and explanation

**Summary Of The Paper:**

This paper focuses on the out-of-distribution detection task and proposes a new method called CIDER, which trains the model by optimizing a dispersion loss and a compactness loss. The compactness loss encourages samples to be close to their class prototypes while the dispersion loss encourages large angular distance among different class prototypes. This paper also investigate EMA for updating prototype parameters. Experiments demonstrate the effectiveness of CIDER.

**Summary Of The Review:**

This paper makes a detailed analysis to OOD detection problem and proposes many metrics (dispersion, compactness and separability) for evaluating OOD performance, which is impressive. Experiment results are thorough, which is helpful for understanding CIDER and its effectiveness compared to previous methods. I just list some points that can be further improved.

1. Updating class prototypes during training seems a little weird. In Algorithm 1, the class prototype $\mu_c$ is updated using EMA first (line 8) and gradient descent w.r.t. Eq. (7) latter (line 12). Why adopting such a two-stage scheme and what if updating prototypes by optimizing Eq. (7) directly without EMA?\

2. Figure 3: It will be better to compare with contrastive loss and show OOD embeddings, just like Figure 1 in KNN+ paper.

---

### Official Review · Reviewer_Vej6 · 2022-10-25

**Confidence:** 4
**Correctness:** 4
**Technical Novelty And Significance:** 3
**Empirical Novelty And Significance:** 4
**Recommendation:** 8

**Clarity, Quality, Novelty And Reproducibility:**

The paper is overall clearly written, besides the point that is discussed in the weakness section. The paper is excellent in quality, particularly regarding the extensive empirical study. The proposed method may not be completely novel but it seems to be new in the OOD detection context. The paper provides enough information to reproduce its results.


**Strength And Weaknesses:**

## Strength

* The proposed method is simple and significant. This finding seems to be generalizable to multiple different applications.
* The paper is overall clearly written with meticulously described details.
* The paper provides extensive empirical study with highly convincing results.

## Weaknesses & Questions

* One thing that is not very clear is how exactly the class prototypes ($\mu$'s) are updated. The reason of the confusion is that there are two equations that can update the class. Both the dispersion loss (Eq.(6)) and the exponential moving average (Eq.(8)) govern the update of the prototypes. Looking at Algorithm 1 in the Appendix, it seems like both equations are applied sequentially. Please correct me if I am wrong.
    * If it is correct that the two equations are both applied to update the prototypes, I wonder why we need Eq.(8). Wouldn't the embedded vectors $z$ eventually cluster around the corresponding prototype by simply enforcing the compactness loss, Eq.(5)? I see that Figure 6(b) shows the effect of this moving average update on the final performance but do not understand why this even affects the quality of the final representations.
* The biggest weakness of this paper is that the finding of the paper is mostly empirical. It would be nicer to have a deeper interpretation regarding the dispersion loss, which is the key contribution of the paper, so that we can have a distilled insight on representation learning and OOD generalization.

**Summary Of The Paper:**

This paper proposes a new out-of-distribution detection method that is based on supervised contrastive learning in a hyperspherical space.  The core component of the proposed method is a novel regularizer that encourages the separation of class centroids. The resulting hyperspherical space is highly effective in distance-based OOD detection methods.

**Summary Of The Review:**

This paper presents solid results on improving OOD detection on hyperspherical representation space.

---

### Official Review · Reviewer_Fpyq · 2022-10-25

**Confidence:** 5
**Clarity, Quality, Novelty And Reproducibility:** The paper is clearly written and the …
**Correctness:** 3
**Technical Novelty And Significance:** 3
**Empirical Novelty And Significance:** 3
**Recommendation:** 6

**Strength And Weaknesses:**

Strengths:

1. Exploiting hyperspherical embeddings for OOD detection is an interesting and effective idea.

2. The proposed dispersion loss is shown to be effective for OOD detection and improving ID classification accuracy.

3. Extensive experiments are conducted to evaluate various aspects of the approach.

a. Proposed approach can be combined with existing parametric (Mahalanobis) and non-parametric (KNN+) approaches (Table 2).
b. The dispersion loss improves both OOD detection and ID classification accuracy (Table 3).
c. The proposed approach improves class separation as expected.
d. Effective for hard OOD samples (CIFAR 10 vs CIFAR 100)

Weaknesses:

1. Authors claim that the OOD samples lie between ID classes. How to validate this claim beyond the visualizations? In some case, OOD sample can be close to ID clusters as well. How does the proposed approach handle such situations? Please refer to the following
Kaur, R., Jha, S., Roy, A., Park, S., Sokolsky, O., & Lee, I. Detecting oods as datapoints with high uncertainty. ICML Workshop on Uncertainty and Robustness in Deep Learning, 2021.

2. Is the proposed loss functions particularly effective while considering the hyperspherical embeddings or can be used with standard euclidean embedding as well?

3. How do various values of the parameter $\lambda$ in Eq.7 affect the OOD detection and ID classification performance?

4. Does the approach require retraining the backbone encoder network with pre-trained networks by just learning the final projection head as shown in Fig. 1?

**Summary Of The Paper:**

This paper presents an approach to detecting out-of-distribution (OOD) samples using hyperspherical embeddings. Specifically, two loss functions: dispersion loss and compactness loss are proposed to increase the inter-class distance and decrease the intra-class distance between samples, respectively. The approach is evaluated on benchmark datasets and various ablation studies are conducted to justify various components of the approach. The results are impressive.

**Summary Of The Review:**

Even though the idea of maximizing the inter-class distance and minimizing the intra-class distance is explored in various contexts, applying this with hyperspherical embeddings for OOD detection is novel. The approach is shown to be effective by extensive experiments. Please address the comments in the weaknesses section.

---

### Decision · Program_Chairs · 2023-01-20

**Decision:**

Accept: poster

**Justification For Why Not Higher Score:**

The negatives of the paper are that there's limited understanding developed about the proposed loss terms and some claims that are hard to verify or might be tautological. Like if the observed classes have a loss to maximize the angle between the observed classes on sphere, it almost seems like by definition the out-distribution classes have to lie in between.

**Justification For Why Not Lower Score:**

The positive of the paper is that the results look good overall and the technique looks simple.

**Metareview: Summary, Strengths And Weaknesses:**

The goal of this paper is to develop out of distribution detection tools using hyperspherical embeddings. They achieve this goal by training the embeddings to maximize the likelihood of the labels and a dispersion loss to make the angular distance between prototypes for each class large. The positives of the paper is that the results look good overall and the technique looks simple. The negatives of the paper are that there's limited understanding developed about the proposed loss terms and some claims that are hard to verify or might be tautological. Like if the observed classes have a loss to maximize the angle between the observed classes on sphere, it almost seems like by definition the out-distribution classes have to lie in between. The reviewers are all positive about this paper (some increased their score after the author rebuttal).

**Note From Pc:**

if the above contains the word "oral" or "spotlight" please see: "oral" presentation means -> notable-top-5% and "spotlight" means -> notable-top-25%. As stated in our emails, we are disassociating presentation type from AC recommendations